# SIGNED BINARIZATION : UNLOCKING EFFICIENCY THROUGH REPETITION-SPARSITY TRADE-OFF

## ABSTRACT

Efficient inference of Deep Neural Networks (DNNs) on resource-constrained edge devices is essential. Quantization and sparsity are key algorithmic techniques that translate to repetition and sparsity within tensors at the hardware-software interface. This paper introduces the concept of repetition-sparsity trade-off that helps explain computational efficiency during inference. We propose Signed Binarization, a unified co-design framework that synergistically integrates hardware-software systems, quantization functions, and representation learning techniques to address this trade-off. Our results demonstrate that Signed Binarization is more accurate than binarization with the same number of non-zero weights. Detailed analysis indicates that signed binarization generates a smaller distribution of effectual (non-zero) parameters nested within a larger distribution of total parameters, both of the same type, for a DNN block. Finally, our approach achieves a 26% speedup on real hardware, doubles energy efficiency, and reduces density by 2.8x compared to binary methods for ResNet 18, presenting an alternative solution for deploying efficient models in resource-limited environments.

## 1 INTRODUCTION

Despite significant strides in accuracy, the burgeoning complexity and resource demands of deep learning models pose challenges for their widespread adoption across a wide range of domains (He et al., 2016; Brown et al., 2020; Graves et al., 2013; Jain et al., 2022; Mandlekar et al., 2021). This requires the development of innovative techniques to enhance DNN efficiency during inference on edge devices. Two such techniques have been studied extensively: binarization and sparsity. Binarization, a form of quantization, results in weight repetition as only two values appear repeatedly in the weight tensor (Courbariaux et al., 2015). This approach significantly trims the memory footprint of the weight tensor, thereby decreasing memory I/O during inference (Hegde et al., 2018). In contrast, sparsity leads to zero weight values. Since anything multiplied by zero is zero, weight sparsity leads to *ineffectual* multiplications (Wu et al., 2021). This approach reduces memory I/O during inference by not reading activations that would be multiplied by zero weights (Gong et al., 2020). Thus, both these techniques are geared towards reducing memory I/O during inference.

DNN inference efficiency is usually achieved by leveraging either binarization or sparsity. Ternary introduces sparsity to binary by introducing an additional zero weight in conjunction to binary weights. Ternary was conceived with the reasonable assumption that transitioning from binary to ternary models would only minimally impact inference latency due to the effect of zero weights (Li et al., 2016). However, advances in the contemporary hardware-software systems have revealed a substantial increase in latency during such transitions (Prabhakar et al., 2021; Fu et al., 2022).

This paper introduces the concept of the repetition-sparsity trade-off, a phenomenon that explains the inference efficiency of binary and ternary weight quantization. A traditional binary network chooses maximization of weight repetition while being ignorant of weight sparsity, whereas a ternary network introduces weight sparsity at the expense of weight repetition. For instance, transitioning from binary to ternary networks increases the number of possible unique 3x3 2D filters from 512 ($2^9$) to 19683 ($3^9$). This makes it exponentially harder to extract efficiency using filter repetition.

The conventional approach to deploying efficient DNNs has largely been a two-stage process: training the DNN with binarization or ternarization (Bai et al., 2018), followed by formulating a system design that reduces memory I/O by leveraging both weight repetition and/or sparsity (Hegde et al.,

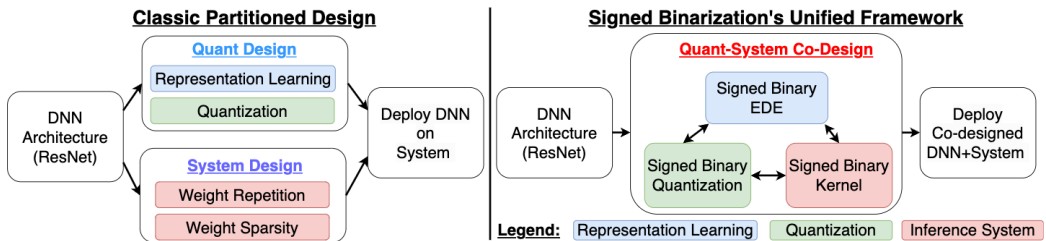

Figure 1: **On the left:** The conventional, isolated approach where hardware-software systems and quantization methods are designed separately leading to inefficient inference. **On the right:** The Signed Binarization approach, a unified design framework, performs quantization-system co-design to address the repetition-sparsity trade-off, thereby enhancing computational efficiency. Extended diagram in Supplementary A.

2018). This partitioned design approach has revealed a pitfall: increased memory I/O during inference due to the repetition-sparsity trade-off. However, recent advancements in DNN inference systems can now exploit repetition and sparsity to improve efficiency as shown in (Prabhakar et al., 2021; Fu et al., 2022). To this end, this paper proposes an elegant, full-stack, quantization-system co-design framework called Signed Binarization . The foundational concept of this framework centers on local binarization, which ultimately results in global ternarization. It is designed with a deep understanding of the repetition-sparsity trade-off. Signed Binarization re-imagines different levels of stack working together in synergy to enhance computational efficiency during inference while retaining the model's accuracy: a) co-design with SOTA systems, called signed binary kernels, that exploit both repetition and sparsity during DNN inference; b) it introduces signed binary quantization functions that cause sparsity without decreasing repetition; c) it offers insights for representation learning with signed binary EDE that influences latent full-precision weights during training to improve accuracy. This allows us to reap significant benefits of exploiting sparsity during DNN inference, as shown in (Wang, 2020; Chen et al., 2019), while retaining benefits caused by repetition.

In this paper, we present the Signed Binarization framework to improve the efficiency of DNN inference by identifying and addressing the trade-off between repetition and sparsity. Thereby, encouraging a collaborative design between different layers of the computational stack to enhance efficiency. Compared to binary, signed-binary co-design improves energy efficiency, reduces latency, and enhance the performance of DNN inference, pushing the Pareto front by providing higher accuracy when both have comparable number of effectual (non-zero valued) parameters as seen in Figure 2. Based on our analysis and empirical evidence, we envision that the Signed Binarization framework has potential to positively shape the evolution of more efficient and high-performing deep learning models.

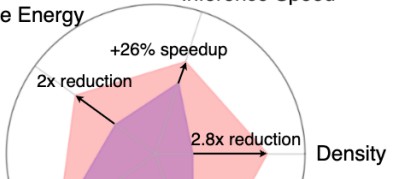

Figure 2: **Binary vs. Signed-Binary ResNet on ImageNet**: The pronounced spread indicates that signed binarization comprehensively outperforms traditional binary methods in various aspects. It ensures competitive accuracy and pushes Pareto front, exhibiting a +2.5% improvement when both methods employ the comparable number of effectual (non-zero) parameters. Moreover, our method enhances inference efficiency, achieving a 26% speedup, doubling energy efficiency, and reducing density by 2.8x for the same backbone.

## 2 BACKGROUND

**Quantization** Quantization in deep learning involves mapping of real-valued numbers to a select set of discrete values to streamline and optimize computations. The Binary Quantization technique assigns any real-valued number to either +1 or -1 (Courbariaux et al., 2015). The primary objective behind this is to simplify operations in DL hardware, turning multiply-and-accumulates (MACs) into mere accumulate operations (Rastegari et al., 2016). Conversely, Ternary

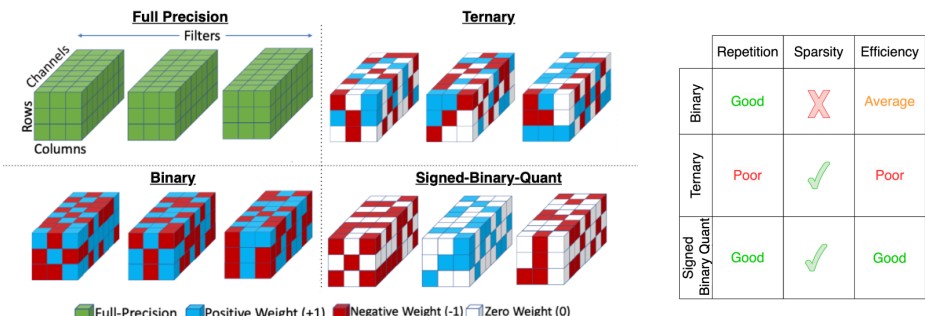

Figure 3: **Concept Diagram on the left:** The diagram shows the comparison of Binary, Ternary, and Signed Binary Quantization in terms of visual representation of their quantized weights. **Qualitative Evaluation on the right:** The table qualitatively evaluates them in terms of weight sparsity, weight repetition, and inference efficiency.

Quantization designates values to +1, -1, or 0 (Li et al., 2016). This determination is based on a threshold, $\Delta$. Values exceeding $\Delta$ are assigned +1, those below receive -1, while others become zero. For both quantization methods, the output values can undergo scaling using a factor, $\alpha$ (Bulat & Tzimiropoulos, 2019; Bulat et al., 2019).

**Weight Sparsity**   Weight Sparsity in DNN implies that there is repetition of weight with a value equal to zero. The basic idea is that since $0 \times x = 0$ for any real-valued scalar $x$, if the weight is zero, the multiplication is *ineffectual* and should be skipped (Gong et al., 2020). Sparsity in weights is static during inference (Dai et al., 2020). Therefore, if the value of weight is zero, we can choose not to load activations corresponding to that weight (Qin et al., 2020). This can lead to a reduction in data movement, memory accesses, and MACs thereby reducing computations and hence resulting in efficient DNN inference. This approach has been effective on ASICs and general-purpose devices (Hegde et al., 2019; Wang et al., 2021; Dai et al., 2020; Gong et al., 2020).

**Weight Repetition**   Extended background in supplementary C. Quantization of weights leads to the same value being repeated again and again in the weight tensor. This phenomenon is known as weight repetition (Hegde et al., 2018; Sze et al., 2020). Since the weights are fixed during DNN inference (Dai et al., 2020), this leads to opportunities for improving efficiency during inference with respect to time and energy by exploiting the repetition of weights and reducing memory accesses (Sze et al., 2020). This concept was first introduced in BNN (Courbariaux et al., 2016) that talked about filter-repetition and demonstrated in UCNN (Hegde et al., 2018) that improved efficiency on ASICs by reordering the weights and thus reorders activations to reduce arithmetic operations. Recent work (Prabhakar et al., 2021; Fu et al., 2022) extends this idea even further by using both repetition and sparsity for efficient inference.

## 3 REPETITION-SPARSITY TRADE-OFF

Consider a convolutional layer with typical 3×3 filters. Binary models, which have $2^9$ unique filters, aim for efficiency by maximizing weight repetition and ignoring sparsity. In contrast, ternary models employ $3^9$ unique filters, introducing sparsity for increased expressivity (Li et al., 2016) but inadvertently causing an exponential decrease in repetition. Recent works (Hegde et al., 2018; Prabhakar et al., 2021; Fu et al., 2022) show a significant slowdown in inference when using ternary when compared with binary due to decreased repetition. This extends to bit-utilization; binary systems can potentially be represented by a single bit, whereas ternary systems require two bits due to the inclusion of zero-valued weights. Moreover, recent findings indicate that runtime doubles when transitioning from one to two bits during the deployment of ResNet18 on ARM CPUs, showcasing the influence of bit-width distinction on inference time (Cowan et al., 2020). These differences can be encapsulated by what we call the repetition-sparsity trade-off, that is crucial during inference of these models. Sign-Binarization aims to address this trade-off to achieve efficient inference while retaining model accuracy. It aims to maintain the number of unique 3x3 filters using two's complement and utilize only one bit during inference, aspiring to combine the merits of both repetition and sparsity for efficient inference while retaining competitive accuracy.

## 4 SIGNED BINARIZATION

The objective is to create a quantization-system co-design method that acknowledges the trade-off between repetition & sparsity by performing local binarization, ultimately resulting in global ternarization.

### 4.1 PROBLEM FORMULATION

Conventional k-bit quantization schemes offer $2^k$ unique weight choices per element, creating a dense setup. Conversely, the (k+1)-bit scheme provides a sparse setup with $2^k + 1$ choices, introducing zero weight and generating ineffectual multiplications. This disparity places the two schemes at opposite ends of the repetition-sparsity spectrum. The proposed signed-binary framework seeks to bridge this divide by maintaining the k bit-width while adding sparsity and preserving weight repetition. It does this by replacing one of the $2^k$ weights with zero, which forms $2^k$ unique quantization functions, fostering k-bit quantization. This trade-off, most prominent at k=1, translates the dense scheme to binary and the sparse to ternary quantization, yielding signed-binary options with quantization function value sets {1,0} and {0,-1}. Acting as a gradient, this strategy mimics binary quantization locally while aligning with ternary quantization globally, paving the way for signed-binary's quantization-software system co-design, wherein the region of signed binary quantization is designed collaboratively with the signed binary kernel to minimize memory I/O, thereby boosting the efficiency over traditional binary networks.

Let the convolutional layer have a $R \times S$ kernel size and $C$ input channels and has $K$ such filters. Let $C_t$ the tile size in $C$ dimension, i.e., a sub-dimension of $C$, effectively partitioning the filter into smaller blocks. The quantization function takes latent full-precision weights $\mathbf{W}$ as input and outputs the quantized weight $\mathbf{W}^{quant}$. We define the region to be sign-binarized as $R \times S \times C_t$ where $C_t \leq C$. The quantized weight $\mathbf{W}^{quant}$ would be the product of the sign-factor $\beta$ and the bitmap $\mathbf{U}$.

$$\mathbf{Q} : \mathbf{W} \rightarrow \mathbf{W}^{quant}; \quad \mathbf{W}^{quant} = \beta \mathbf{U} \tag{1}$$

$$\forall \mathbf{W} \in \mathbb{R}^{R \times S \times C_t}; \quad \beta \in \{+1, -1\}; \quad \mathbf{U} \in \{0, 1\}^{R \times S \times C_t} \tag{2}$$

### 4.2 METHOD

Developing an efficient signed-binarization framework requires a redesign due to its distinct nature from existing methods, as a latent FP weight can be assigned at any of $2^k$ unique signed-quantization functions. The challenges arise from the need to design signed-binary quantization functions, that enable the model to learn meaningful representations. Signed-binary quantization must be compatible with signed-binary kernels to observe real improvement in efficiency. Local binarization techniques must accommodate a global ternarization strategy, which presents additional challenges such as determining suitable regions for local binarization and assigning these regions to their designated signed binary quantization functions. Finally, we need to explore techniques to fine-tune latent full-precision weights during the training phase to bolster model accuracy.

In response, our approach is outlined through systematic ablation studies on ResNets trained on the CIFAR-10 dataset, summarized in Tables 1-5 along with inference in Section 6.1. The elements in grey represent our default configurations for the method. (For in-depth insights into our experimental setup and implementation details, please refer to the supplementary section F). The ensuing sections delve deeper into each of the aforementioned components and challenges, offering a comprehensive view of our method's evolution and its adaptations to various configurations.

#### 4.2.1 EFFICIENT INFERENCE WITH CO-DESIGN

**Co-designing Efficient Signed Binary Kernel**: To observe efficiency improvement, the signed binary kernel needs to be attuned to the repetition-sparsity trade-off. Our approach is to co-design a signed binary kernel with signed binary quantization to fully exploit both repetition and sparsity during inference. The key insight is that recent work (Fu et al., 2022; Prabhakar et al., 2021; Hegde et al., 2018) splits the dot products of input and weights into smaller blocks via tiling to improve data locality during inference. We use SumMerge (Prabhakar et al., 2021) as the signed binary kernel such that the signed binary quantization's size $C_t$ is a multiple of signed binary kernel's tile size, i.e., $C_t = \max(C, kC')\forall k \in \mathbb{Z}^+$ and $C' =$ signed binary kernel's tile size. This allows a single processing step of the kernel to see one signed binary quantization function leading to efficiency.

| Arch | FP | T | B | SB |
|---|---|---|---|---|
| ResNet20 | 92.10 | 90.86 | **90.20** | 90.05 |
| ResNet32 | 92.90 | 92.03 | 91.51 | **91.55** |
| ResNet44 | 93.30 | 92.40 | 91.93 | **91.98** |
| ResNet56 | 93.63 | 92.90 | 92.42 | **92.52** |
| ResNet110 | 93.83 | 93.33 | 92.64 | **92.68** |

Table 1: Binary and Signed-Binary achieve comp--arable accuracy on CIFAR-10 ablations.

| $\%\{0,1\}$ filters | $\%\{0,-1\}$ filters | Acc |
|---|---|---|
| 0 | 1 | 88.84 |
| 0.25 | 0.75 | 89.32 |
| 0.5 | 0.5 | **90.05** |
| 0.75 | 0.25 | 89.30 |
| 1 | 0 | 89.07 |

Table 2: Equal percentage of positive and negative signed-binary regions leads to best model accuracy.

| Signed Binary EDE | Acc |
|---|---|
| Disabled | 88.4 |
| Enabled | **88.7** |

Table 3: Signed Binary EDE improves accuracy.

| Signed Binary Region | Acc |
|---|---|
| $C_t = C$ | **88.6** |
| $C_t = C/2$ | 87.9 |

Table 4: Variation of $C_t$ leads to competitive accuracy.

| Signed Binary $\Delta$ | Acc |
|---|---|
| $0.01 \times \max|\mathbf{W}|$ | 90.01 |
| $0.05 \times \max|\mathbf{W}|$ | **90.05** |

Table 5: Signed-Binary is not sensitive to the choice of $\Delta$.

**Ablations for Signed-Binary**: We conduct experiments by training ResNets on CIFAR-10 dataset for the method. Tables 2-5 use ResNet20 trained on CIFAR10. The default configuration for our method is marked in grey .

### 4.2.2 Designing Signed Binary Quantization Functions

**Signed Binary Quant Functions**: Our method involves the strategic use of two distinct quantization functions. In this section, we intend to meticulously design these functions, characterized by the value sets $\{0,1\}$ where the sign-factor $\beta_1$ is equal to 1, and $\{0,-1\}$ where the sign-factor $\beta_{-1}$ is -1. The scaling factor $\alpha_i$ mirrors $\beta_i$ where $i = \pm 1$. Following (Zhu et al., 2016), we define the threshold value as $\Delta = 0.05 \times \max(|\mathbf{W}|)$. To assess the efficacy and sensitivity of the chosen thresholds, we experiment with different $\Delta$s using signed-binary quantization (see Section 4.2.2) to find stable performance across various configurations, as depicted in Table 5. Functions are defined as:

$$\mathbf{W}^{quant} = \left\{ \begin{array}{ll} \alpha_1 & \text{if } \mathbf{W} \geq \Delta \text{ and } \beta = 1 \\ \alpha_{-1} & \text{if } \mathbf{W} \leq -\Delta \text{ and } \beta = -1 \\ 0 & \text{Otherwise} \end{array} \right\} \tag{3}$$

$$\frac{\partial L}{\partial \mathbf{W}} = \left\{ \begin{array}{ll} \alpha_1 \times \frac{\partial L}{\partial \mathbf{W}^{quant}} & \text{if } \mathbf{W} > \Delta \text{ and } \beta = 1 \\ -\alpha_{-1} \times \frac{\partial L}{\partial \mathbf{W}^{quant}} & \text{if } \mathbf{W} < -\Delta \text{ and } \beta = -1 \\ 1 \times \frac{\partial L}{\partial \mathbf{W}^{quant}} & \text{Otherwise} \end{array} \right\} \tag{4}$$

**Intra-Filter Signed-Binary Quant** In this section, we delve deeper into signed binary quantization by introducing variations in $C_t$ with respect to the constant value $C$, aiming to identify the optimal setting for $C_t$. We design this approach to emphasize the changes in performance across different thresholds. Table 4 assesses the impact on representation quality by adjusting the $C_t$ values during the training. The result shows that intra-filter signed binary co-design preserve a competitive level of representation quality even with a reduced $C_t$ and setting of $C_t = C$ works best.

**Inter-Filter Signed-Binary Quant** Building on intra-filter, where $C_t = C$, this is the simplest signed binary quantization, and we term it as Inter-Filter Signed binary quantization. In essence, this method involves assigning each filter to one of two distinct signed binary quantization functions, enabling an efficient representation as $C_t = C$. We contrast it with binary and ternary quantization schemes in an apples-to-apples comparison. Table 1 illustrates ResNets of varying depths trained using different quantization schemes. The signed-binary quantization maintains comparable accuracy against traditional quantization approaches across different architectures.

**Value Assignment of Signed-Binary Quant Functions** To fine-tune our network's performance, it's crucial to examine how different value assignments within filters affect accuracy. This experiment focuses on studying the effects of varying the proportions of filters with $\{0,1\}$ and $\{0,-1\}$

value assignments. As illustrated in Table 2, we explore the delicate balance between positive and negative binary values to find the best way to optimize network performance. The data indicates that having an equal mix of these values is more beneficial, clearly outperforming the methods that use only a single type of sparse quantized function. This observation underscores the importance of using both positive and negative binary regions to achieve effective learning of representations.

### 4.2.3 LEARNING BETTER REPRESENTATIONS

**Signed Binary EDE during Backward Propagation:** Binary networks typically exhibit a prominent peak of latent full-precision weights at zero, a consequence of employing a sign function. This leads to fluctuations in latent weight assignments and ensuing instability, leading to degrading the model's accuracy. To counteract this, EDE gradually approximates the sign function's derivative during the backward propagation phase, enhancing both the update capability and gradient accuracy (Qin et al., 2021). In contrast, signed-binary co-design abstains from using zero-value thresholding for local binarization, manifesting instead two distinct peaks at non-zero values of $\pm\Delta$ (see Figure 5b). This necessitates the development of a new method to prevent gradient information loss during backward propagation for such networks. Responding to these challenges, we introduce Signed binary EDE, a method specifically tailored to stabilize the fluctuations of latent full-precision weights in sign-binary around $\Delta = \pm 0.05 \max(\mathbf{W})$, thereby fostering improved representations as delineated in Table 3.

Signed binary EDE operates by first determining $t$ and $k$ using the equations $t = T_{\min}10^{\frac{i}{N} \times \log \frac{T_{\max}}{T_{\min}}}$, $k = \max\left(\frac{1}{t}, 1\right)$ where $i$ is the current epoch and $N$ is the total number of epochs, $T_{\min} = 10^{-1}$ and $T_{\max} = 10^1$. Subsequently, we adapt $g'(\cdot)$ for sign-binary: $g'(x) = kt\left(1 - \tanh^2(t(x \pm \Delta))\right)$, to compute the gradients with respect to $w$ as $\frac{\partial L}{\partial \mathbf{w}} = \frac{\partial L}{\partial Q_w(\mathbf{w})} g'(\mathbf{w})$

## 5 SIGNED BINARY CO-DESIGN PUSHES THE PARETO-FRONTIER

In this section, we evaluate the efficacy of signed binary in preserving the quality of learned representations when transitioning from the conventional binarization technique. Initially, we train ResNets on the CIFAR10 and ImageNet datasets using signed binary, and benchmark the results against existing *state-of-the-art* methods in subsection 5.1. Additionally, subsection 5.2 examines the latent full-precision and quantized weights in detail to provide a holistic understanding of the retained representational capacity of the trained model.

### 5.1 COMPARISON WITH OTHER WORKS

To ascertain the true potential of signed binary codesign, we compare it with existing SOTA methods. We benchmark on CIFAR10 and ImageNet datasets by training ResNet-{20,32} and ResNet-{18,34} models. Weight Sparsity leads to effectual and ineffectual multiplications caused by non-zero and zero-valued parameters respectively (Wu et al., 2021). Conversely, model parameters are of two types, effectual (or non-zero valued) parameters and ineffectual (or zero valued) parameters. Since only effectual parameters lead to computation during inference, we compare different binary methods on two axes: model accuracy on the Y-axis and effectual binary model parameters on

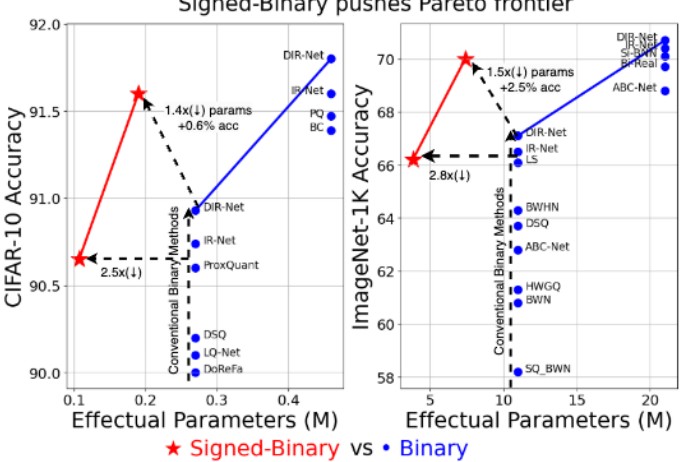

Figure 4: Comparison of Signed binary and conventional binary methods on CIFAR10 and ImageNet datasets. Signed binary pushes the Pareto frontier, providing superior accuracy with respect to effectual parameters and exhibiting a significant reduction in effectual parameters of equivalent models.

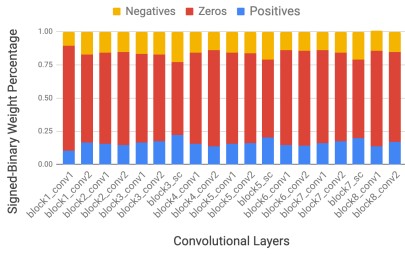

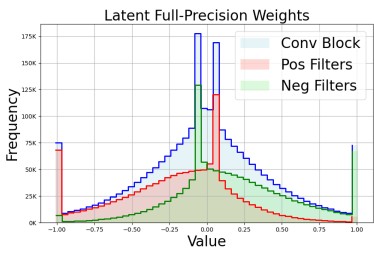

(a) **Quantized Weights**  (b) **Latent Full Precision Weights**

Figure 5: **On the left: Quantized Weights**, are depicted, illustrating a distribution reminiscent of ternary networks but with weights distinctively segregated across filters, enhancing weight-repetition and inference efficiency. **On the right: Latent Full Precision Weights** in a signed-binary conv block maintain a *blue distribution* akin to binary's Laplacian. In sign-binary, *total parameters* can be divided between *positive* and *negative* signed-binary filters. These parameters can be subdivided into non-zero valued effectual and zero-valued ineffectual ones. Notably, while the total parameter's looks like binary's zero-mean Laplace distribution, individual filters do not. The effectual and ineffectual parameters' *green-red* and *red-green* distributions *resemble* Laplace and Gaussian distributions, respectively. Sign Binary reduces computations during inference when compared to binary, as it reduces effectual parameters from *blue* to *green-red* distribution.

the X-axis (refer to supplementary F for details and baselines). As shown in Figure 4, signed-binary exhibits Pareto optimality against state-of-the-art methods. It achieves a $+0.7\%$ and $+2.5\%$ increase in accuracy on the CIFAR10 and ImageNet datasets, respectively, coupled with $1.4\times$ and $1.5\times$ reduction in effectual parameters respectively. Concurrently, it realizes approximately a $\sim 2.5\times$ and $\sim 2.8\times$ decrease in effectual parameters for equivalent models on the respective datasets. These results emphasize the potential of signed binary co-design in creating efficient and accurate models.

## 5.2 VISUALIZING LATENT FP AND QUANTIZED WEIGHTS

**Latent FP weights.** Binarization quantization leads to a zero-mean Laplacian distribution of latent full-precision weights (Xu et al., 2021). ResNet18 trained on Imagenet in Figure 5, we observe that despite the pronounced sparsity introduced by signed binary quantization, the distribution of latent FP weights across an entire convolutional block resembles a zero-mean Laplacian distribution. Signed-binary filters are neither zero mean, nor exhibit a Laplacian distribution (supplementary I). This shows that even if individual filters don't align with a zero-mean or Laplacian distribution, collectively, the convolution block resembles these characteristics. Moreover, the presence of four distinctive peaks overlaying Laplacian resembling distribution for full-precision latent weights are due to: (a) Two peaks at the extremes appear because of clamped weights at +1 and -1, and (b) two intermediate peaks arising from the employment of threshold functions defined at $\pm\Delta$, analogous to zero-valued peak observed in binary network due to the use of sign quant function.

**Quantized weights.** We investigate the distribution in quantized weights of a trained signed-binary model by plotting the percentage distribution of quantized signed-binary convolutional blocks in ResNet18 trained on Imagenet. Figure 5 reveals a roughly consistent, equal proportion of both positive and negative weights. This observation of a roughly equal proportion of positive and negative weights is also observed in ternary quantization (Zhu et al., 2016). However, a crucial distinction arises in the distribution of positive and negative weights within a convolutional layer. Like ternary, both positive and negative valued weights are present within a layer. However, signed binary co-design *bifurcates* them across different filters. This design decision improves inference efficiency.

## 6 SIGNED-BINARY CO-DESIGN IMPROVES INFERENCE EFFICIENCY

In this section, we aim to demonstrate that switching from binary to signed binary leads to an increase in inference efficiency. First, we demonstrate that signed-binary is more efficient than binary and ternary because it acknowledges the repetition-sparsity trade-off. Next, we understand how the presence of weight sparsity in a DNN leads to benefits during DNN inference. For these experiments, we focus on signed-binary ResNet18 trained on ImageNet.

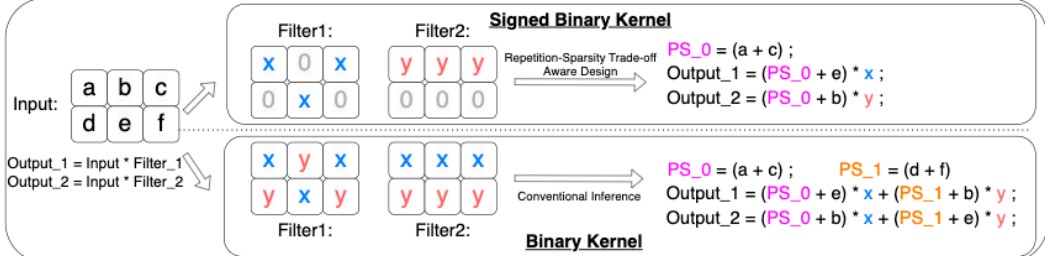

Figure 6: **Signed Binary leads to efficient inference as it acknowledges repetition-sparsity trade-off through quantization-system co-design**: Visualizing inference in Section 6.1 when using recent systems Prabhakar et al. (2021); Fu et al. (2022). Weight repetition enables binary models to skip work by re-using partial sums within and across filters. Signed-Binary, takes this even further by reducing the number of effectual operations by leveraging sparsity while retaining repetition. (Step by step explanation in Supp B)

## 6.1 EXPLOITING REPETITION & SPARSITY WITH SIGNED BINARY

We posit that the performance of binary and ternary weight quantization methods in inference may be hindered due to their neglect of the crucial repetition-sparsity trade-off. Weight sparsity allows us to utilize upcoming sparse tensor operation technology to accelerate inference by skipping zeros during inference (Wu et al., 2021). Concurrently, exploiting weight repetition leads to diminishing arithmetic operations and data movement (Hegde et al., 2018). However, traditional binary networks prioritize the maximization of weight repetition, overlooking the potential benefits of weight sparsity, while ternary networks induce weight sparsity but compromise on weight repetition. In contrast, we hypothesize that signed binary co-design, being attuned to this trade-off, promises enhanced efficiency in actual device inference. To validate our hypothesis, we deploy quantized ResNet-18 models on Intel CPUs, rigorously measuring inference times under varying conditions.

**Experimental Setup and Methodology**    We use SumMerge (Prabhakar et al., 2021) for performing inference of quantized and sparse DNNs on Intel CPUs (details in the supplementary D). All experiments are conducted under identical test environments and methodologies. Within our experiments utilizing SumMerge, we explore two distinct configurations: (1) with sparsity support deactivated, the software does not distinguish between zero and non-zero weights, relying solely on weight repetition; (2) with sparsity support activated, the software additionally omits computations involving zero weights. We provide detailed analysis into both per-layer speedups and the aggregate speedup across different quantization strategies relative to binary quantization.

**Result and Analysis**    Please see Figures 6 and 7. We observe in Figure 7 that signed-binary is consistently more efficient than binary and ternary for every quantized layer and the model overall by 1.26x and 1.75x faster respectively, when exploiting both repetition and sparsity.

This result can be explained as follows: *(A) When sparsity support is turned off*: The software is only relying on repeating values within the weight tensor for speedup. Because binary and signed-binary have two unique values per convolutional filter, they take similar time for DNN inference. Ternary is much slower as it has three unique values per convolution filter which makes extracting efficiency by using weight repetition exponentially harder. *(B) When sparsity support is turned on*: The software not only cares about the repeating values in the weight tensor but also skips computations on zero weights to improve the runtime. Here we observe that ternary is slower than binary, because the reduction in work due to sparsity is not able to compensate for the exponential decrease in weight repetition. Our method, on the other hand, does not suffer from this problem and is able to exploit weight repetition and weight sparsity to the fullest and is most efficient.

## 6.2 UNDERSTANDING BENEFITS OF SPARSITY DURING INFERENCE

In this section, we aim to understand the potential benefits of weight sparsity to a binary network during inference. To do this, we count the number of quantized weights with zero values and divide it by the total number of quantized weights to calculate the percentage of sparsity. We find that

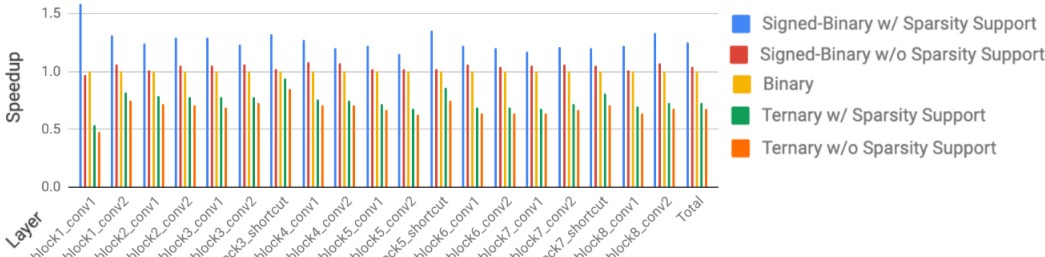

Figure 7: **Efficiency Analysis wrt Binary ResNet18 on Intel CPU**: Our study shows signed-binary excelling in every convolutional layer, depicted by bars for **signed-binary**: w/ sparsity support and w/o sparsity support ; **ternary**: w/ sparsity support and w/o sparsity support ; and one for **binary** , indicating its 100% density. Please refer to Figure 5. The inference computations for signed-binary w/o sparsity support is aligned with *blue* distribution, matching binary performance. On the other hand, sparsity support reduces computation to *green*-*red* distribution. Because *negative* and *positive* valued quantized weights exist in separate regions of the network by design (see Figure 3), signed-binary quantization retains repetition, resulting in speedup.

signed-binary ResNet-18 trained on ImageNet has 65% sparsity. Since density is (1 - sparsity) (Wu et al., 2021), ResNet-18 has 35% density (see Figure 5a). if we switch from conventional binary to signed-binary, we decrease the density from 100% to 35%. We would like to leverage the low density to reduce the amount of computation activity during inference:

**Throughput**    In a model with low density, represented by $1/x$, there is one effectual multiplication for every $x$ total multiplications. By eliminating the ineffectual computations and the time associated with them, there is the potential to improve throughput by a factor of $x$ (Emer et al., 2021). Given that the signed-binary has a 35% aggregate density—meaning only 35% of the multiplications are effectual—exploiting sparsity during inference can lead to a potential $2.86\times$ increase in throughput compared to dense binary $\{1,-1\}$. In practice, the speedup we observe on real hardware is in the range 1.26x-1.75x as the support for unstructured sparsity on general-purpose devices is an active area of research (Wu et al., 2021). This speedup is comparable to the speedup due to unstructured sparsity on general-purpose devices in recent papers (Gong et al., 2020; Hegde et al., 2019).

**Energy Reduction**    To estimate energy reduction due to the unstructured sparsity during inference, we use the cycle-level micro-architectural simulator (Muñoz-Matrínez et al., 2021) of a sparsity supporting ASIC (Qin et al., 2020). We take their publicly released code and use it under default configuration (details in supplementary D). We observe that decreasing density from 100% to 35% for one-bit ResNet 18 leads to a ~2x reduction in energy during inference. Thus, switching from binary to signed-binary would lead to significant improvements in power consumption on ASICs.

## 7    DISCUSSION AND FUTURE WORK

The paper introduces the concept of repetition-sparsity trade-off and proposes Signed Binary, a quantization-system co-design framework that addresses this trade-off by performing local binarization resulting in global ternarization. Signed Binary exhibit Pareto optimality compared to prior binary methods, improving accuracy per effectual parameter and enhancing computational efficiency during inference with respect to speed, energy, and density. However, signed binary requires training from scratch, which can be time-consuming and computationally expensive. Additionally, the under-studied repercussions of quantization and, consequently, signed binary quantization on model bias warrant further exploration. This work focuses on quantized system co-design for standard backbones to address repetition sparsity trade-off. While specialized low-bit specific architectures (Guo et al., 2022; Zhang et al., 2022) for signed binary are outside the scope of this co-design exploration, we encourage future research to explore these architectures for signed binary. Despite these limitations, Signed Binary presents a promising approach for training efficient models that are both accurate and computationally efficient, particularly well-suited for applications where hardware resources are limited, such as mobile devices and embedded systems.

## 8 REPRODUCIBILITY STATEMENT

We provide all the hyperparameters for the key experiments including instructions on how to train the models in the supplementary E. Further, since our work is about quantization system co-design, we provide all the hyperparameters and configurations to reproduce our inference experiments in supplementary D. Finally, implementation details and baselines for our method can be found in supplementary F.

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

# A  METHOD VISUALIZATION

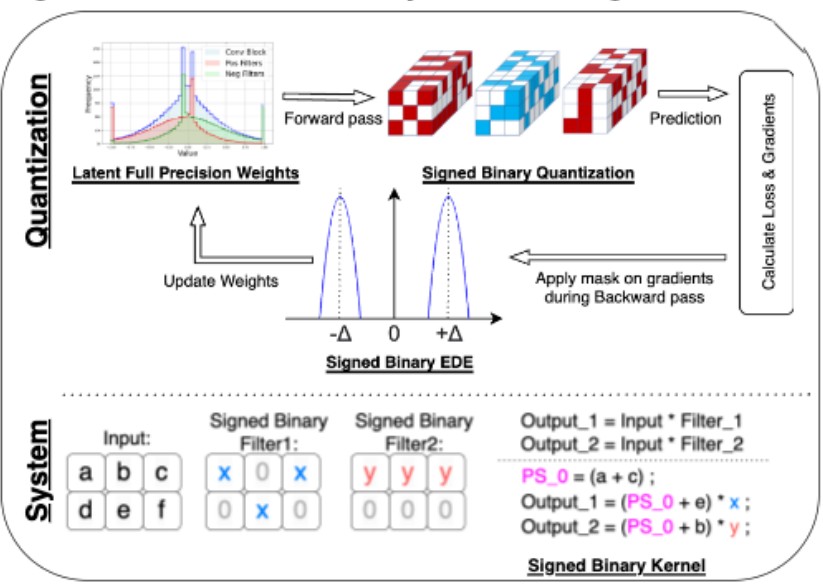

Figure 8: The unified co-design framework, known as the Signed Binary approach, effectively integrates all components to address the repetition-sparsity trade-off, thereby enhancing computational efficiency while maintaining competitive accuracy. Signed binary EDE leads to more meaningful representations, Signed binary quantization balances repetition and sparsity, and Latent-Full-Precision weights align to create a smaller Laplacian-like distribution, as shown in the green-red color (as opposed to the blue-colored Laplacian-like distribution that constitutes for total parameters similar to binary's Laplacian). Finally, on the bottom, co-design with signed binary kernel leads to reducing arithmetic operations, resulting in faster inference on real hardware.

# B  VISUALIZING EXPLOITATION OF REPETITION AND SPARSITY

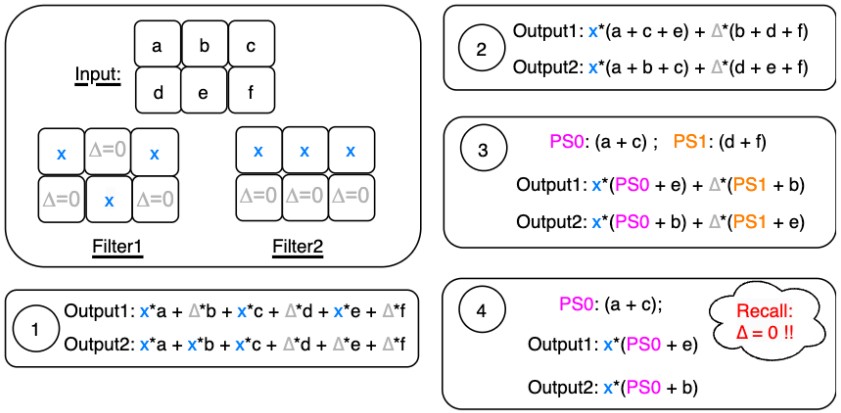

Figure 9: Visualization of repetition and sparsity during inference in modern systems: (1) Input and two filters that need to be multified (2) naive multiplication of weights and activations to create output 1 and output 2. (3) Re-ordering of weights and activations to simplify work within each filter (and exploit intra filter repetition phenomenon (Hegde et al., 2018)) (3) Using partial sums to reduce the work across filters (and exploit intra filter repetition phenomenon (Hegde et al., 2018)) (4) exploiting sparsity (by acknowledging repetition of zero weights as a special case of weight repetition (Sze et al., 2020; Hegde et al., 2018; Prabhakar et al., 2021).

## C  EXTENDED BACKGROUND

**Weight Repetition**: Quantization of weights in DNN leads to the same value being repeated again and again in the weight tensor. This phenomenon is known as weight repetition (Hegde et al., 2018). Since the weights are fixed during DNN inference, this leads to opportunities for optimizations. The objective is to improve efficiency during inference with respect to time and energy by exploiting the repetition of weights and reducing memory accesses (Sze et al., 2020).

Using weight repetition for efficiency first originated in the BNN (Courbariaux et al., 2016) which talked about the idea of filter repetition in BNNs for efficient inference. The authors highlight that the number of unique filters possible in a binary setting is bounded by the filter size. This is explained with an example that a 3x3 2D filter can only have $2^9$ unique filters in a binary network. (Courbariaux et al., 2016) states that there are only 42% of unique filters per layer on average which can lead to reducing the number of XNOR operations by 3x.

UCNN (Hegde et al., 2018) was the first work that demonstrated efficiency by using weight repetition on ASICs. It performed efficient inference by reordering the weights and thus reorders activations and operations. This reduces memory access and decreases the number of arithmetic operations required during DNN inference. For example, if the filter weights are $[a, b, a, a]$ and activations are $[w, x, y, z]$, UCNN would reorder it as $a \times (w + y + z) + b \times (x)$ for efficient inference (Sze et al., 2020). They show that weight repetition is widespread and abundant across a range of networks (like ResNet, and AlexNet) trained on a variety of datasets (CIFAR10, ImageNet). Finally, they show that weight repetition leads to a 4x improvement in throughput normalized energy consumption with only 24% area overhead relative to the same baseline.

SumMerge (Prabhakar et al., 2021) extends the idea of UCNN even further by using both weight repetition and weight sparsity for efficient DNN inference. For example, if $b = 0$ in the previous example, SumMerge would compute $a \times (w + y + z)$ during inference. It enabled repetition and sparsity aware inference on general purpose devices and showed that arithmetic operations could be reduced by 20x and 15x for binary and ternary networks respectively. Q-Gym (Fu et al., 2022) is Meta's latest inference system that treats weight repetition as a combinatorial optimization problem to achieve SOTA speedup on CPUs and GPUs for both single-threaded and multi-threaded settings.

## D  INFERENCE EXPERIMENT SETUP

**Deploying on CPUs**  We use SumMerge (Prabhakar et al., 2021) for this task. We run all experiments on Intel Xeon Gold 6226 CPU. In order to make our test environment as close as possible to the test environment of the authors of (Prabhakar et al., 2021), we disable simultaneous multi-threading, and enable 2MB huge pages and disable dynamic frequency scaling as well. The test methodology is exactly the same as used by the authors of (Prabhakar et al., 2021), i.e., each experiment is run 50 times when the machine is unloaded and the values for run with the lowest execution time are reported. All arithmetic operations are in floating-point. All DNN inference experiments are subject to identical test environments and methodology.

**ASIC**  We use STONNE (Muñoz-Matrínez et al., 2021), a cycle-level microarchitectural simulator for DNN Inference Accelerator SIGMA (Qin et al., 2020) for this experiment. We use the docker image released by the authors of (Muñoz-Matrínez et al., 2021). We use the standard configuration of SIGMA with 256 multiplier switches, 256 read ports in SDMemory and 256 write ports in SDMemory. The reduction networks is set to ASNETWORK and the memory controller is set to SIGMA_SPARSE_GEMM. We use SimulatedConv2d function in the PyTorch frontend version of STONNE. For a given convolutional layer, we run STONNE twice, once with 0% sparsity and once with 65% sparsity. We calculate the reduction in energy consumption by dividing the energy of the dense convolutional layer by the energy of the sparse convolutional layer. Since the weights' precision (or bit-width) is a parameter of SIGMA, the reduction in energy due to sparsity when compared to the dense model is not a function of the precision of the weights of the DNN.

## E  TRAINING EXPERIMENTAL SETUP

**CIFAR10**  The data loader pipeline consists of simple augmentations - padding by 4 pixels on each size, random crop to $32 \times 32$, Random Horizontal Flip with probability 0.5, and normalization. We

train from scratch for 350 epochs and use the Adam Optimizer (Kingma & Ba, 2014). We start with an initial learning rate of 0.01 and reduce it by a factor of 10 at epochs 150, 200, and 320. For apples-to-apples comparison with binary and ternary, we do a sweep over batch sizes {16, 32, 64, 128, 256} and activation functions (ReLU, PReLU, TanH) and report the best top-1 validation accuracy. For ablations on (1) value assignment percentage and (2) comparison with binary networks with comparable effectual operations, we select the batch size the to be 32 and activation function to be PReLU. We compare the binary and signed-binary methods on their non-zero valued parameters of quantized layers as other aspects would be similar. When comparing against prior art works, we run these methods on our setup and report numbers. Further ablations on batch size and activation function provided in the appendix.

**ImageNet**   We train ResNet-18 (He et al., 2016) using SBWN on ImageNet (Deng et al., 2009). We use standard practices used to train binary networks like (1) normalize the input using batch-norm (Ioffe & Szegedy, 2015) before convolution instead of after convolution (Rastegari et al., 2016), (2) the first and the last layers are not quantized (Zhu et al., 2016; Pouransari et al., 2020; Li et al., 2016; Bai et al., 2018). We use a first-order polynomial learning-rate annealing schedule with Adam optimizer (Kingma & Ba, 2014) along with PReLU (He et al., 2015; Maas et al., 2013). We use FFCV dataloader (Leclerc et al., 2022) with simple augmentations - Random Resize Crop to $224 \times 224$, Random Horizontal Flipping and Color Jitter with (brightness, contrast, saturation, hue) set as (0.4, 0.4, 0.4, 0). We decrease the learning rate from $2.0 \times e^{-4}$ to $2.0 \times e^{-8}$ while training for 320 epochs and do not use weight decay and batch size of 256 for training. We compare the binary and signed-binary methods on their non-zero valued parameters of quantized layers as other aspects would be similar. When comparing against prior art, we report numbers as is from the literature due to the high compute cost of running these experiments. Furthermore, for ResNet 34, we simply increase the model size.

## F   BASELINES AND IMPLEMENTATION

**Baselines for Comparison with Prior work**: Figure 4 shows comparison against prior-art methods: DIR-Net Qin et al. (2023), LQ-Net Zhang et al. (2018), DSQ Gong et al. (2019), BC Courbariaux et al. (2015), ProxQuant Bai et al. (2018), IR-Net Qin et al. (2021), DoReFA Zhou et al. (2016), SQ-BWN Dong et al. (2017), BWN Rastegari et al. (2016), ABC-Net Lin et al. (2017), BWHN Hu et al. (2018), LS Pouransari et al. (2020), Bi-Real Liu et al. (2018), .

**Implementation**   Signed-Binary quantization is a local binarization scheme, i.e., the quantization function takes full-precision latent weights of region of a convolutional layer as input and maps it to either $\{0,1\}^{R \times S \times C_t}$ or $\{0,-1\}^{R \times S \times C_t}$. The values of the quantization function for these prede-termined regions of a convolutional layer are determined randomly before training commences and remain unchanged. Different regions can have distinct quantization functions for a convolutional layer. This framework categorizes these regions into two buckets, optimizing efficiency by grouping them based on their quantization function values for more streamlined training. For example, if $C_t = C$, signed binary quantization becomes a pre-filter quantization scheme and each filter will have a different quantization function. We quantize the full-precision latent weights of a convolu-tional layer from $\mathbb{R}^{\{R \times S \times C \times K\}}$ to $\{0,1\}^{R \times S \times C \times K \times P}$ and $\{0,-1\}^{R \times S \times C \times K \times (1-P)}$ where P is the percentage of filters whose quantization functions have the values $\{0,1\}$ such that $K \times P$ is an integer.

## G   SIGNED-BINARY VS BINARY WRT EFFECTUAL PARAMETERS

We would like to compare binary with signed-binary when the DNN has the same number of non-zero parameters. Signed-Binary ResNet trained on CIFAR10 has slightly greater than 50% sparsity. If we reduce the total number of parameters of the binary ResNet by half, the resulting model would have comparable number of non-zero weights to signed-binary ResNet. This is done by reducing depth (see Table 6a) and by reducing width (see Table 6b). We train these models under identical conditions (setup details in the appendix). To clarify, row 1 & row 2 of Table 6 have the same number of total parameters while row 1 & row 3 have a comparable number of non-zero parameters. Thus, signed-binary leads to a higher accuracy than binary when both methods have comparable number of effectual operations.

| Quant | # Parameters | Depth | Acc |
|-------|--------------|-------|-----|
| SB | 0.46M | 32 | 91.55% |
| B | 0.46M | 32 | 91.22% |
| B | 0.27M | 20 | 90.16% |

(a) **Reducing number of parameters by reducing depth**: We observe that accuracy of binary is 1.3% lower than signed-binary with comparable non-zero weights.

| Quant | # Parameters | Width | Acc |
|-------|--------------|-------|-----|
| SB | 0.27M | $1\times$ | 90.05% |
| B | 0.27M | $1\times$ | 90.20% |
| B | 0.14M | $\lceil 0.7\times \rceil$ | 88.5% |

(b) **Reducing number of parameters by reducing width**: We observe that accuracy of binary is 1.7% lower than signed-binary with comparable non-zero weights.

Table 6: **Binary vs Signed-Binary with comparable non-zero weights**: We observe that Signed-Binary achieves higher accuracy when compared to binary with comparable effectual operations.

## H    ADDITIONAL ABLATIONS ON CIFAR-10

We perform ablation on (1) batch sizes, (2) non-linearity on CIFAR10 (Krizhevsky & Hinton, 2009) dataset and report the numbers in Table 7a and 7b respectively. The setup is the same as mentioned above. We observe that for our method, there is a drop in accuracy with higher batch size, PReLU (Maas et al., 2013) works best and it is not sensitive to the choice of delta.

| Batch Size | Accuracy (Top-1) |
|------------|------------------|
| 16 | 89.44 |
| 32 | 90.05 |
| 64 | 89.62 |
| 128 | 89.59 |

(a) **Ablation on Batch Size**: The setup is identical across batch sizes and the non-linearity used is PReLU. We observe a decrease in accuracy when a high batch size of 256 is used.

| Non-Linearity | Accuracy (Top-1) |
|---------------|------------------|
| ReLU | 88.64 |
| PReLU | 90.05 |
| TanH | 88.75 |
| LReLU | 89.22 |

(b) **Ablation on Non-Linearity**: The setup is identical across non-linearity and the batch size used is 32. We observe that PReLU works best for our method.

Table 7: **Additional Ablations on CIFAR10**: Abations on batch size and non-linearity for SB.

## I    LATENT FULL-PRECISION WEIGHTS & STANDARDIZATION

Binary quantization has shown to improve accuracy from standardization of latent full-precision weights. However, this trend is not observed in signed-binary networks.

| Standardization Strategy | Accuracy (%) |
|--------------------------|--------------|
| Local Signed-Binary Regions | 59.1 |
| Global Signed-Binary Block | 61.2 |
| No Standardization | 61.4 |

Table 8: Comparison of Standardization Strategies on Accuracy

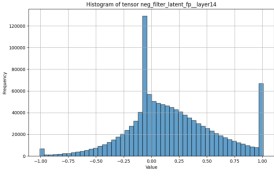

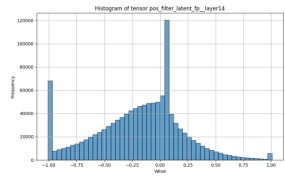

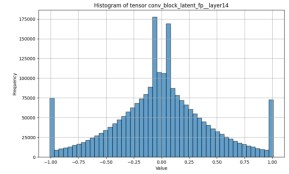

(a) Negative SB Filters          (b) Positive SB Filters.          (c) SB Conv Block.

Figure 10: Distribution of Latent Full-Precision Weights: While the local signed-binary weights are neither zero mean, nor Laplacian, the entire conv block is zero mean Laplacian distribution. The peaks in the image are because of clipping at $\pm 1$ along with thresholding at $\pm\Delta$.

## J    ARITHMETIC REDUCTION ABLATION

Figure 11 compares the arithmetic reduction of binary, ternary, and signed-binary quantization for convolutional layers of various sizes. The reduction is measured relative to dense computation. The figure is created using Sum-Merge's DNN inference algorithm, which generates data-flow graphs with weight-dependent structures. The experiment is conducted using the original test settings and methodologies as described by the original authors. The reported arithmetic operation reduction is for uniform distribution tensors. We observe that signed-binary is the most efficient and achieves the maximum arithmetic reduction for all convolutional layers.

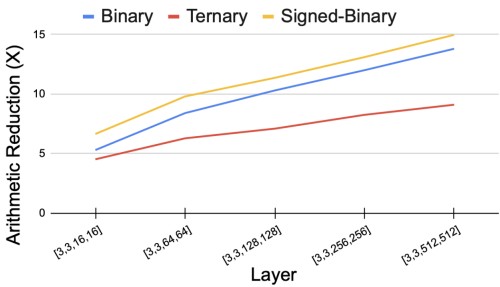

Figure 11: Arithmetic Reduction (higher is better) for Binary, Ternary and Signed-Binary. Results are relative to dense computation.

## K    DATASETS

Licenses of ImageNet (Deng et al., 2009) and CIFAR10 (Krizhevsky & Hinton, 2009) datasets used in this paper are listed in Table 1. Every accuracy reported in this paper is on validation set of the dataset.

| Dataset | License | Source |
|---------|---------|--------|
| ImageNet | Non-Commercial | ILSVRC2012 |
| CIFAR10 | N/A | CIFAR |

Table 9: **Dataset with Licenses**: License and source of the datasets used.

ImageNet and CIFAR10 are standard publicly used datasets. Since they do not own their images, therefore they do not have a release license. Actual images may have their own copyrights but ImageNet provides stipulations for using their dataset (for non-commercial use). We do not recommend using the resulting signed-binary models trained on these datasets for any commercial use.

