# OpenReview forum: "Signed-Binarization: Unlocking Efficiency Through Repetition-Sparsity Trade-Off"
_ICLR.cc/2024/Conference — Submitted to ICLR 2024_

### Official Review · Reviewer_pHUD · 2023-10-29

**Soundness:** 3 good
**Presentation:** 2 fair
**Contribution:** 2 fair
**Rating:** 5
**Confidence:** 3

**Summary:**

This paper proposes a new quantization method called Signed Binarization for training efficient deep neural networks. The key idea is performing local binarization of network weights, which results in global ternarization. This exploits the repetition-sparsity tradeoff to improve efficiency during inference. The method uses two distinct quantization functions and alternates between them locally within the network filters.

**Strengths:**

1. The proposed methods appear to be easily reproducible.
2. The concept of a repetition-sparsity tradeoff in quantized networks is insightful and well-motivated.

**Weaknesses:**

1. The contribution of this paper is limited. This paper seems like an incremental work over prior binary network research.
2. The paper should be carefully proofread. There are some typos, e.g., “This necessitate creating …” and “aiming to address …”. I also feel a little bit confused about Table 6, because it doesn’t have a table, and seems like just an overall caption of Table 1 to Table 5. The models used in some Tables are missing, e.g., Table 2 to Table 5.
3. The performance is limited as signed-binary is 1.26x and 1.75x faster than binary and ternary respectively.

**Questions:**

1. Figure 4 seems unclear to me. The authors mentioned they compared against 13 prior methods, but there are less than 13 blue dots on both subfigures. Also, what are the models used in Figure 4, Figure 5, and Figure 6?
2. Signed Binarization alternates quantization functions locally, but how are the regions/filters assigned to each function? Is this predetermined or optimized during training? Could this assignment influence the results?
3. What is the difference between Signed Binary method without sparsity support and Binary method?

---

> ### Author Response · Authors · 2023-11-18
> **Rebuttal Response**
>
> We thank the reviewer for the detailed evaluation and positive comments. The following clarifications are inline with the updated version of the paper.
> ___
> **Summary1 “paper proposes a new quantization method…”**
>
> Thank you for review. Please refer to **common response 1**. We believe the confusion lies in thinking Signed Binary is just {0,±1}. The paper proposed the idea of repetition-sparsity trade-off (**Section3**) and performs quantization-system co-design (**Figures1 and 8**) to create signed-binarization to address the trade-off (**Abstract**).  SB = {Signed Binary Quantization + Signed Binary EDE + Signed Binary Kernel}.
>
> **Summary2 Signed Binary exploits the repetition-sparsity tradeoff**
>
> We agree with the reviewer. However, the causality is flipped. Repetition-Sparsity tradeoff causes Signed-Binary to be faster. Repetition-Sparsity tradeoff is the key insight, signed-binary is a tool to prove it.
> ___
> **Q1 Diff between Signed Binary w/o Sparsity support and Binary {1,-1}?**
>
> Thank you for this question. We want to reiterate that Signed-Binary is a quant-system co-design approach. For a conceptual difference, contrast **Figures 3 & 8**. For quantitative difference, ablations in **Tables 1&7 and Supplementary G** show that both have competitive accuracy wrt total parameters and signed-binary is more accurate wrt effectual parameters
>
> **Q2 Diff between Signed Binary Quantization Functions {0,±1} and Binary {1,0} w/o Sparsity support?**
>
> Thank you for this question. The difference is the accuracy. We trained ResNet18-ImageNet using the two quant schemes in an ablation. {0,±1} was 6% more accurate than {1,0}.
>
> **Q3 Paper is incremental work over prior binary research**
>
> We request the reviewer to please cite prior work so that we can highlight our contributions better. We believe that our contributions are substantial and are re-iterated in **Common Responses 3, 4, and 5**.
>
> **Q4 Confusion about primary contribution**
>
> We thank the reviewer for the question. Please refer to **common responses 3 and 4**. We hope this addresses all the concerns about the primary contributions.
>
> **Q5 performance is limited as signed-binary is 1.26x faster than binary**
>
> We understand the reviewer’s perspective. However, as shown in **Figure 2**, we report gains on Energy, Throughput, Latency, Density, and Accuracy wrt Effectual Params. Our gains are not individualistic but combined because of quantization-system co-design. Also, please refer to **Common Response 5**.
>
> If you meant to ask why Signed binary is 1.26x faster than binary and 1.75x faster than ternary while giving significant improvements in every other domain – please refer to **Section 6.2 Para 2** for a detailed explanation. Sparsity support for general-purpose devices is an active area of research and our speedup is comparable with recent work. To put this in perspective, **[1]** demonstrates the real speedup of 1.28x-1.64x on general-purpose devices due to unstructured sparsity. Prior works, like ternary quant works (ProxQuant ICLR), simply do not deploy their quantization schemes on general-purpose devices. To our knowledge, this is because the first system to support ternary on CPUs was published in 2021! To our knowledge, we are the first ML work to deploy the proposed method on real hardware. This is a strength of this work.
>
> Please note: To our knowledge, the first work to deploy ternary quant works
>
> **Q6 Tables**
>
> *Table 2-5*: ResNet 20 + CIFAR10
>
> *Figure 5-6*: ResNet 18 + ImageNet
>
> *Figure 4*: [ResNet18 + ImageNet, ResNet34 + ImageNet, ResNet20 + CIFAR10, ResNet32 + CIFAR10]. Thank you for pointing this out. We have reported the numbers for all baselines in whichever dataset they were available. We have changed Figure 4 and text to highlight this clearly. For example: ProxQuant (ICLR’19) only reported results on CIFAR10 and subsequent literature reviews don’t report ProxQuant on ImageNet.
>
> **Q7 Signed Binary Training and Implementation**
>
> Thank you for this question. This is mentioned in **supplementary F**. However, we will clarify this here in the text: Filters are assigned to each function before training and are predetermined. Additionally, **Tables 2 & 4** provide ablations quantization function assignments during finding the optimal design.
>
> signed-binary quantization functions should not be swapped during training as it can influence the results. Please refer to **Section 5.2 Para 2 & Figure 10**. This is because the distributions of {1,0} and {-1,0} are neither zero-mean nor laplacian! They need to be combined to create Laplacian & Gaussian-like distributions for effectual and ineffectual params respectively. We stabilize these distributions using Signed-Binary-EDE.
> ___
>
> [1] Gong, Zhangxiaowen, et al. "Save: Sparsity-aware vector engine for accelerating dnn training and inference on cpus." In 2020 53rd Annual IEEE/ACM International Symposium on Microarchitecture (MICRO),

---

> ### Author Response · Authors · 2023-11-23
> **Last few days of the discussion period**
>
> As this phase of the discussion period concludes, we wanted to reach out to see if there are any remaining questions or clarifications we can provide!
>
> The key concerns that Reviewer pHUD pointed out were (1) clarification about models used; (2)  figures and tables; and (3) Clarification on the speedup achieved against the landscape of SOTA inference systems. We have updated figures and tables and marked the corresponding models used in each one. Additionally, figures 1 and 8 help provide more information about the intention of this work. We have also provided explanations with references to demonstrate that the speedup is meaningful. We believe the remaining inline responses, extra experiments, and revised draft (uploaded) also work to address any remaining concerns.
>
> We're excited to see how our reviewers view our paper in light of these changes. Again, if there are any remaining questions, please let us know as we will stay available. Thanks so much!

---

### Official Review · Reviewer_ZM5v · 2023-10-29

**Soundness:** 2 fair
**Presentation:** 2 fair
**Contribution:** 2 fair
**Rating:** 3
**Confidence:** 4

**Summary:**

This paper proposes a new binarization scheme which uses {-1, 0} binarization for some tiles and {0, +1} binarization for the others. It aims to improve the so-called "repetition-sparsity trade-off" an the accuracy-model size Pareto frontier of binarized neural networks. The paper also visualize model effective parameters and speedup analysis on Intel CPU.

**Strengths:**

This paper has a good structure. The figures are nice.

**Weaknesses:**

1. The paper conveys confusing messages on several basic concepts.
    * About weight “repetition” in BNNs (Section 2 last paragraph). The efficiency of BNNs does not mainly come from the “repetition” of 1-bit filters so that redundant memory access can be reduced. It comes from the reduction in bitwidth compared to high-precision floating-point numbers so that more efficient matmuls can be applied. The motivation of the paper is therefore ill-posed.
    * Questionable claim in Section 3: “Binary models aim for efficiency by maximizing weight repetition”. There are no specific constraints or regularizations on the objective function to make a binarized CNN optimized for repeating its filters. The paper needs to provide evidence to show that BNNs are intentionally learning repetitive weights.

2. The goal of improving the repetition-sparsity trade-off is questionable. Repetition and sparsity do not seem to be a trade-off, i.e., improving one doesn’t compromise the other. For example, in the context of the paper, a model with zero weights is extremely sparse and in the meantime has extremely repetitive filters.

3. The method described in Section 4 seems to be incomplete. It uses {-1, 0} binarization for some weight tiles and {0, +1} binarization for the others. This means that on hardware, 0 and 1 will correspond to different values for different tiles. What will be the overhead when accumulating the output of those tiles? The paper needs to provide an analysis.

4. There are limited baseline methods. The paper claims it pushes the Pareto frontier, but in Figure 4 it does not compare to the recent advancements in BNNs, e.g., [1], [2], etc.
    * [1] N. Guo et al., Join the High Accuracy Club on ImageNet with A Binary Neural Network Ticket, arxiv’22.
    * [2] Y. Zhang et al., PokeBNN: A Binary Pursuit of Lightweight Accuracy, CVPR’22.

**Questions:**

Questions are included in the weakness section above.

---

> ### Author Response · Authors · 2023-11-18
> **Extended background and rebuttal (1/2)**
>
> We thank the reviewer for the detailed evaluation and comments. We thank the reviewer for giving his perspective. We would like to thank the review as this review led us to the improvement of the manuscript as we have added Figure 6, Figure 9, and Figure 11. Additionally, we have added Supplementary Sections B, C, and J.
>
> Before answering the raised concerns, we would be thankful if the reviewer gives us a chance to first establish common terminologies. Recent fundamental breakthroughs [2,4,5] in the inference of these models have changed the underlying landscape of efficient processing of DNNs. We use the following background information in answering concerns raised by the reviewer.
> ___
> ___
> **Extended Background** (Supplementary C) :
> ___
> ___
> **B1 What is Weight Repetition and highlight its impact on inference (used in explaining B2)**
>
> Lightning Talk: https://youtu.be/7r9zGOHYf2g?si=wEKq17eTTt7Cda_7
>
> | Year | Method | Claim | Citation |
> | --- | --- | --- | --- |
> | 2016 | Filter Repetition based | 3x reduction in Ops | Section 3.3 of [1] |
> | 2018 | Indirection Table based | 4x reduction in Energy | Abstract of [2] |
> | 2020 | Section in Textbook | Listed as fundamental method | Page 177 of [3] |
> | 2021 | Graph Based | 20x reduction in Ops | Figure 10 of [4] |
> | 2022 | Combinatorial Optimization Based | SOTA on CPU, GPU, Single & Multi Threaded | Abstract of [5] |
>
> **B2 Filter Repetition vs Weight Repetition (used in answering Q2)**
>
>
> Filter repetition is one technique for exploiting weight repetition, which is a broader term. Recent methods [4,5] care about the number of unique weights.
>
> **B3 Quantization enforces Weight Repetition (used in answering Q3)**
> | Citation | Claim |
> |-|-|
> | Figure 3 of [2] | Weight repetition is widespread and abundant across a range of networks of various sizes and depths for quantized models |
> | Page #153 (line 7 ) & #177 of [3] | A. quantization of the weights can also be thought of as a form of weight sharing, as it reduces the number of unique weights |
> | | B. Repeated weights appear in a filter due to quantization as discussed on Page 153 |
> | | C. Weight repetition reduces the number of weight accesses from memory as well as the number of multiplications |
> | Figure 10 & 11 of [5] | Weight repetition gives SOTA performance for a wide range of networks. |
>
> **B4 Comparing energy consumed for ADD/Multiply and data movement (used in answering Q1)**
>
> The reduction of data movement from the DRAM is the biggest bottleneck (for latency, energy, and throughput) for efficient processing of DNNs [3]. To put this in perspective, loading two numbers from DRAM is 6400x (206x) more expensive than adding (multiplying) them [3]. Thus the goal of efficient inference is summarized as reducing data movement [6]. Quantization makes things efficient because it reduces the amount of data that needs to be read (Chapter 7 of [3]).
> ___
> [1] Binarized Neural Networks, NeurIPS 2016 (https://arxiv.org/abs/1602.02830)
>
> [2] UCNN, ISCA 2018 (https://dl.acm.org/doi/10.1109/ISCA.2018.00062)
>
> [3] Vivienne Sze, Joel S. Emer - Efficient Processing of Deep Neural Networks-Morgan and Company (2020) ISBN: 9781681738314
>
> [4] SumMerge ICS 2021 (https://dl.acm.org/doi/pdf/10.1145/3447818.3460375)
>
> [5] Q-Gym PACT’22 (https://dl.acm.org/doi/10.1145/3559009.3569673)
>
> [6] Eyeriss Tutorial (https://eyeriss.mit.edu/tutorial.html)
>
> [7] N. Guo, arxiv’22.
>
> [8] PokeBNN CVPR’22.
> ___
> ___
> **Rebuttal Starting**
> ___
> Using the aforementioned fundamentals, we would be thankful if the reviewer gave us a chance to provide clarifications.
> ___
> **Summary1 “The paper proposes a new binarization scheme"**
>
> Thank you for your review. Please refer to **CommonResponse 1**. This work is not a new conventional binarization scheme. Instead, we propose the repetition-sparsity trade-off along with Signed-Binarization, a quantization-system co-design framework to address this trade-off.
>
> **Summary2 “It aims to improve the accuracy-model size Pareto frontier”**
>
> We feel that there has been a misunderstanding. We talk about the accuracy-effectual parameter Pareto front and never discuss the accuracy-model size Pareto front in the paper. Please refer to **Section 5.1**

---

> ### Author Response · Authors · 2023-11-18
> **Continued Rebuttal (2/2)**
>
> **Q1 Efficient matmul is the primary reason for efficiency and not reduced memory accesses due to repetition**
>
> Thank you for the question. We feel that there has been a misunderstanding: the term “weight repetition” is the equivalent of weight quantization during the processing of DNNs on hardware (see **B3**). The major factor in efficient inference is achieved by reducing data movement from memory and not by simpler matmuls (see **B4**). To that end, reducing the bit-width and weight repetition (albeit used interchangeably in [3]) have the same effect: reducing data movement from DRAM. Prior work (see **B1**) shows that repetition reduces memory access up to 20x is plays a central role in SOTA system design [5].
>
> **Q2 Explain Claim: Binary chooses to maximize repetition of weights**
>
> Binary does not need constraints or regularizations on the objective function. Since binary has two unique weights, repetition happens due to the pigeonhole principle (**B3, B2, Page 2 Line 1 of [3]**). Repetition has always existed but was missed by prior works. Signed-Binary identifies this and addresses our newly identified trade-off between repetition and sparsity. See **Common response C1**
>
> **Q3 Provide evidence for Binary learning repetitive weights**
>
> Carrying forward with the previous answer, consistent with prior work (See **B3 above**), we observe a similar trend in **Figure 11** of this work, where we achieve up to a 15x reduction in operations due to weight repetition for ResNet.
>
> **Q4 Show Repetition-Sparsity Trade-Off**
>
> Thank you for the question. Signed binary requires fewer operations than binary (**Figure6**). This can also be shown a higher reduction in operations (**Figure11**) leading to faster inference (**Figure7**). This makes SB more efficient than B (**Figure2**). The table below demonstrates that improving one compromises the other. This makes signed-binary most efficient:
>
> | Where | What | Binary | Signed-Binary| Ternary |
> |-|-|-|-|-|
> | Figure 6 | Toy Example | 8ADD ops | **4 ADD ops** | 6ADD ops |
> | Figure 11 | Ops Reduction for [3,3,64,64] layer | 5.3x | **6.65x** | 4.52x |
> | Figure 7 | Latency | 1x | **1.32x** | 0.54x |
>
> | | Signed Binary | Binary |
> |-|-|-|
> | Energy(↓) | **0.51x** | 1x |
> | Throughput(↑) | **2.86x** | 1x |
> | Density(↓) | **0.35x** | 1x |
> | Latency(↓) | **0.78x** | 1x |
>
> **Q5 "In the context of paper, a model with zero weights is extremely sparse and has extremely repetitive filters"**
>
> Please name the specific model of your concern.
>
> Let’s visualize this using a toy example for  [3,3,256,256] conv layer. This layer has 256 x 256 = 65.5K 3x3 filters. Assuming a uniform distribution of filters, pigeonhole principles will give us:
>
> | | # unique 3x3 filters | Repetition per filter: |
> |-|-|-|
> | Ternary | 19,683 | 3.32x |
> | Binary | 512 | 128x |
> | Signed Binary | 1,024 | 64x |
>
> **Q6 Differentiate between naive {0,1} quant and signed-binary-quant functions {0, ±1}. Demonstrate overhead with analysis**
>
> Thank you for improving our manuscript. We have added **Figures 8 & 9** to visualize inference with signed-binary-quant & naive {0,1} quant. Overhead is only loading *one* additional unique weight {y} per filter in Signed-Binary. Toy example from Fig 9:
>
> *Input*: {a,b,c,d,e,f}
>
> *Signed-Binary Filters*: {x,0,x,0,x,0} {y,y,y,0,0,0}
>
> *Naive {1,0} Filters*: {x,0,x,0,x,0} {x,x,x,0,0,0}
>
> **Inference Dataflow graph Signed-Binary**:
>
> PS0 = (a+c)
>
> Output1 = (PS0 + e)*x
>
> Output1 = (PS0 + b)*y
>
> **Inference Dataflow graph {1,0}**:
>
> PS0 = (a+c)
>
> Output1 = (PS0 + e)*x
>
> Output1 = (PS0 + b)*x
>
> **Analysis with ResNet18/ImageNet**: Signed Binary is 6% more accurate from naive {0,1} while both methods have comparable ops and latency.
> |  | Ops reduction for [3,3,512,512] | Speedup wrt Binary | Accuracy (ablation) |
> |-|-|-|-|
> | {0, ± 1) | 14.51x | 1.25x | 61.94 |
> | (0,1) | 14.61x | 1.29x | 55.23 |
>
> **Q7 Diff between Signed Binary Co-design and [7,8] Binary Model Arch Exploration**
>
> This work explores quantization-system co-design to prove the existence of repetition-sparsity trade-off. This is orthogonal to [7,8] which proposes a custom Conv+Atten-based DNN architecture for binary quantization. While a special Conv+Atten architecture could be designed for signed-binary, our experiments are sufficient to prove the repetition-sparsity trade-off. Finding the best architecture for signed binary is outside the scope of this work. We have added a discussion about  [7,8] in Section 7 of the updated PDF of this work.

---

> > ### Author Response · Authors · 2023-11-23
> > **Last few days of the discussion period**
> >
> > As this phase of the discussion period concludes, we wanted to reach out to see if there are any remaining questions or clarifications we can provide!
> >
> > *The key concerns identified by Reviewer ZM5v were*:
> >
> > (1) Expansion of the background;
> >
> > (2) Clarity on the repetition-sparsity trade-off
> >
> > (3) Analysis comparing signed-binary-quant {0, ±1} with naive {1,0}
> >
> > *We believe that the following revisions address these concerns*:
> >
> > (A) Extended background in the supplementary;
> >
> > (B) New experiments to demonstrate the repetition-sparsity trade-off
> >
> > (C) Demonstration that signed-binary-quant {0, ±1} is more accurate than naive {1,0} without significant latency overhead.
> >
> > Additionally, we have clarified that this work focuses on the accuracy-effectual parameter Pareto front, not the accuracy-model size Pareto front. We believe that the remaining inline responses, additional experiments, and revised draft (uploaded) effectively address any remaining concerns.
> >
> > We're excited to see how our reviewers view our paper in light of these changes. Again, if there are any remaining questions, please let us know as we will stay available. Thanks so much!

---

### Official Review · Reviewer_Ffqa · 2023-10-31

**Soundness:** 3 good
**Presentation:** 2 fair
**Contribution:** 3 good
**Rating:** 6
**Confidence:** 3

**Summary:**

In this paper the authors propose a method to improve the efficiency of binary quantization models by constraining the presence of -1 and 1 values to specific regions within the filter parameters. By enforcing this additional constraint the method achieves locally binary quantization within a region, by allowing values to be either 0 or {-1, 1}, but globally ternary across the entire filter. The result is that within a constrained region values are binarized as well and highly sparsified which reduces the amount of data that must be moved per layer to perform an inference pass. The authors show that this simple application does not greatly impact the accuracy of the resulting network but yields notable performance improvement.

**Strengths:**

- The idea is relatively simple and the description regarding the pros and cons from a performance perspective is made clear in the text.
- Great use of figures (2 and 3) to help the user get the gist of the proposed method. I especially appreciated the color coding and detail provided by Figure 3 to compare and contrast the current work against other approaches.
- While I like the comparisons illustrated in Figure 4 it could be better if there was a way to easily associate the source for each blue dot. For now the reader has to cross-reference the dot with the referenced works to make the comparisons concrete.
- The performance results in Figure 6 clearly demonstrate the increased efficiency of the inference pass for the signed-binary networks compared to normal binary and ternary networks.

**Weaknesses:**

Major:
- My only major concern is with respect to the novelty of the approach. I like the proposed method but it seems to overlap in many ways with previous work, used for instance in post-training quantization methods, to decompose a set of parameters into non-overlapping regions and using a fixed number of values per region. I think of this work as being an extension of that idea where the set of values per region is constrained to come from a very restricted set.

Minor:
- Figure 1 should be expanded upon to include more valuable information or be removed altogether. As it is I don't think it provides a lot of value to the text.
- The proposed method requires retraining from scratch to learn the binarized filters (this was mentioned as a limitation in the text).
- The text does not flow very well in several places in the text and needs a few more passes over the text to clean up awkward sentences.
- Most of the text in the first paragraph of section 5.1 seems to be repeating the same idea regarding effectual and ineffectual values. It's not clear to me that a distinction between effectual and ineffectual parameters vs multiplications necessitates redundancy in the text.
- Though the terminology of "effectual" and "ineffectual" seem to be inherited from prior work I can't say I'm a fan of it. It's not something worth debating but personally, it's not clear to me how much value I gain denoting zeros as "ineffectual".
- Aesthetically speaking I do not care for the wrapped text around Figure 2 or the packed nature of Tables 1-5. Though the tables may fit well together there may be a better way to incorporate Figure 2 by sacrificing Figure 1.

**Questions:**

The method is exclusively applied to convolutional layers, could it be applied more generally to linear layers, possibly decomposed into sets by row or column or some other fixed pattern?

---

> ### Author Response · Authors · 2023-11-18
>
> We thank the reviewer for the detailed evaluation and the comments from the reader’s perspective. The following clarifications are in line with the updated version of the paper.
> ___
> **Q1 Fixed Figure 1**
>
> We thank the reviewer for pointing out the issues in Figure 1. We have now updated **Figures 1 & 8** to show that the proposed method is a quantization-system co-design framework to prove the repetition-sparsity trade-off (and the objective is not to create a new quantization scheme). We hope it now becomes clearer that the paper is about re-visiting design principles, i.e., Signed-Binary = {Signed-Binary Quant + Signed-Binary EDE + Signed Binary Kernel} and not just {0,±1} quantization functions.
>
> **Q2 Method extension to linear layer**
>
> We thank the reviewer for this extensibility idea. Yes, we believe it is possible to extend it to linear layers and have added **Figures 6 & 9** to help the reader visualize the same. Repetition-Sparsity trade-off is independent of the model architecture. Prior work shows that ternary gets worse (read less efficient wrt binary) as models get bigger. We believe this is an exciting extension and implore it to the community to take this up as future work.
>
> **Q3 Ineffectual parameter and multiplication**
>
> We agree with the reviewer and carry forward the terminology of “effectual” and “ineffectual” parameters from previous works [1]. Since zero weights need not be loaded in from the memory they result in an inherent I/O speedup for DNNs. That is why, the difference between “effectual” and “ineffectual” parameters plays an important role in understanding our work.
>
> Ineffectual parameters are the characteristic of the model being processed that signifies how many parameters (read zeros) need not be loaded from the memory. Ineffectual multiplications are characteristic of the underlying system and signify how well the system can take a sparse model and reduce the number of operations due to zero weights.
>
> **Q4 What’s new? Key insight?**
>
> We thank the reviewer for this question. Proposing a new quantization schema, Signed Binary Quant is a subset of our contribution. The other subset of contribution is how it translates to reduction of I/O calls in hardware, this is where the repletion-sparsity tradeoff comes in. Figure 6 shows us that the weight is multiplied after adding all the corresponding activations it should be multiplied with, i.e., x*(PS_0+e). This results in {1,0} quantization schema and {±1,0} quantization schema having near identical inference kernels. The only difference is the non-zero weight value which is being loaded per filter. To the best of our knowledge, no one has put all of these together to provide significant gains in efficiency.
>
> Please refer to **common responses 3, 4, and 5** for findings and impact.
>
> [1] Sparseloop MICRO 2022 (https://sparseloop.mit.edu/)

---

> > ### Author Response · Authors · 2023-11-23
> > **Last few days of the discussion period**
> >
> > As this phase of the discussion period concludes, we wanted to reach out to see if there are any remaining questions or clarifications we can provide!
> >
> > One of the key concerns that Reviewer Ffqa pointed out was the lack of information in Figure 1, which led to confusion over novelty. We believe that our updated Figures 1 and 8, along with the explanation of the intent of this work to demonstrate the repetition-sparsity trade-off, clarify the matter. Additionally, we hope Figures 6 and 9 help the reader visualize how the repetition-sparsity trade-off impacts efficiency. We believe that the remaining inline responses, extra experiments, and revised draft (uploaded) also work to address any remaining concerns.
> >
> > We're excited to see how our reviewers view our paper in light of these changes. Again, if there are any remaining questions, please let us know as we will stay available. Thanks so much!

---

### Author Response · Authors · 2023-11-18
**Common Response**

We thank the reviewers for their insightful comments and valuable feedback. We have uploaded the updated version of the paper with updated Figure 1 (expanded in Figure 8) and added Figure 6 (expanded in Figure 9). The following clarifications are inline with the updated version of the paper:

**C1 Summarize this work. Diff btw Signed Binary Quant and Signed Binary the full-stack framework**

This work proposes the concept of repetition sparsity trade-off (**Section 3**). Prior work designs quantization methods and inference systems separately (**Figure 1**). We propose Signed-Binarization, a full-stack quant-system co-design framework that re-imagines different levels of stack working together (**Figure 8**), namely {Signed Binary EDE, Signed Binary Quant, Signed Binary Kernel} to unlock efficiency (**Figure 2**). This demonstrates the existence of repetition sparsity trade-off (**Abstract, Introduction and Figures 7 & 11**).

**C2 Visualizing inference in SOTA systems**

Unlike prior work, this paper does not use old heuristics to estimate efficiency but rather uses SOTA systems. (**Figure 6**) demonstrates signed-binary and binary filters during inference with SOTA systems. Signed-binary co-design enables skipping more computations leading to faster inference.

**C3 Key insight**

The novel repetition-sparsity trade-off is the key insight of this work. Repetition-sparsity trade-off is a new idea that makes the design decision of {0, ±1} reasonable (which is also new). Signed-binary unlocks efficiency by synergestically co-designing different layers of the stack (signed-binary-quantization, signed-binary ede, signed-binary kernels) (**Figure 1**) to address newly identified repetition-sparsity trade-off. This would not have been possible with binary or ternary quant. The contribution is not simply {0,±1} but how everything fits together due to the Repetition-sparsity trade-off to provide efficiency (**Figure 2**)

**C4 Highlight original findings**

To our knowledge, the following findings are new when compared to prior work:

A) We are the first work to show the existence of a trade-off between repetition and sparsity that impacts efficiency making prior work (both binary and ternary) suboptimal. (**Section 3** )

B) We are the first work to propose Signed Binary Quant {0,±1} (**Section 4.1**). Our co-design makes it more efficient (in energy, latency, throughput, density) than convention {1,-1} (**Figure 2**). We show that Signed Binary Quant {0,±1} learns significantly better representations than Binary Quant wrt effectual parameters (**Section 5.1** ).

C) Unlike Binary and Ternary quantization, we show that Signed Binary Quant {0,±1} does not conventionally learn zero-mean Laplacian distributions for latent full-precision weights. Its individual regions neither exhibit zero mean nor exhibit Laplacian distribution (**Figure 10**). Our Signed Binary EDE stabilizes the weight updates. Finally, we show that despite being distributed in different regions, effectual and ineffectual parameters combine from different regions and create Laplacian-like and Gaussian-like distributions, respectively (**Figure 5b**).

**C5 Impact**

Battery life doubles and throughput can be nearly tripled for edge devices. Model inference is now faster, and reduced density enables the use of bigger models resulting in more accurate performance (**Figure 2**). The holistic improvement due to co-design will make inference significantly more economical.

**C6 Reproducibility Statement**

Added **Section 8**. All hyperparam, setup, and implementation details in supplementary.

---

### Meta-Review · Area_Chair_LuRv · 2023-12-06

**Metareview:**

This paper proposed to improve the efficiency of binary quantization models by constraining the presence of -1 and 1 values to specific regions within the filter parameters. Strengths across the reviews include the simplicity and clarity of the proposed method, and the insightful concept of a repetition-sparsity tradeoff in quantized networks. However, important weaknesses were found by the reviewers as well. A common concern is the novelty: reviewers in general note its overlap with prior work in post-training quantization and binary NN. Incomplete method descriptions were also noted.

During the review, the authors raised a concern about being asked to compare with a concurrent TMLR submission, and they suggested this might imply a potential reviewer COI. AC carefully investigated this issue to conclude there was no COI evidence found. The case was communicated with PCs too.

**Justification For Why Not Higher Score:**

Please see above. While it shows some promise in its approach and results, the issues raised by the reviewers lead to the rejection recommendation.

I've carefully read the authors' rebuttal and noted many clarifications were made to improve the paper's credibility and clarity. However, those will require substantial rewriting to be incorporated into the paper, and might need another round of peer reviews to warrant so.

**Justification For Why Not Lower Score:**

N/A

---

### Decision · Program_Chairs · 2024-01-16

Reject